Brief Communication

# Olfactory misinformation provides refuge to palatable plants from mammalian browsing

Patrick B. Finnerty [1] ✉, Malcolm Possell [1], Peter B. Banks[1], Cristian Gabriel Orlando [1], Catherine J. Price [1], Adrian M. Shrader [2] & Clare McArthur [1] ✉

Mammalian herbivores browse palatable plants of ecological and economical value. Undesirable neighbours can reduce browsing to these plants by providing 'associational refuge', but they can also compete for resources. Here we recreated the informative odour emitted by undesirable plants. We then tested whether this odour could act as virtual neighbours, providing browsing refuge to palatable eucalyptus tree seedlings. We found that protection using this method was equivalent to protection provided by real plants. Palatable seedlings were 17–20 times more likely to be eaten by herbivores without virtual, or real, neighbours. Because many herbivores use plant odour to forage, virtual neighbours could provide a useful practical management approach to help protect valued plants.

Foraging decisions of mammalian herbivores can have costly consequences, devastating areas of habitat restoration and post fire recovery[1,2], and cause billions of dollars of damage in forestry and agriculture[3] globally. Current solutions to problematic herbivory traditionally target removing animals, such as lethal control, or preventing access to plants, such as fencing. These approaches are costly and increasingly limited by practicalities, concerns over animal welfare and non-target ecological effects, so alternative approaches are needed. The most effective alternatives are likely to be those based on understanding and harnessing foraging cues, motivations and decisions[4] of the herbivores.

Generalist mammalian herbivores typically forage by navigating heterogenous landscapes of discontinuous food resources. To maximize foraging efficiency in such landscapes, animals seek high-quality food patches and avoid low-quality patches dominated by unpalatable low-nutrient, chemically defended[5,6] or physically obstructing[7,8] plant species. Palatable plants in such low-quality patches can receive protection against herbivores from their low-quality neighbours—termed associational plant refuge[9,10].

To recognize and select among plants and plant patches, many mammalian herbivores use and rely on plant odour[11,12]. But plant odours are extremely complex. They often comprise hundreds of volatile organic compounds (VOCs), many of which are common among plant species[13] and most likely to be uninformative noise. The VOC information that foraging mammalian herbivores use to recognize plants amidst this complex olfactory noise remains poorly understood.

Defining odour information is crucial to understanding its role in plant–herbivore interactions, in foraging and more broadly in any ecological interactions mediated by odour. It could also be crucial for developing new ways to manage problem browsing[14]. For example, strategically designed artificial odours could be exploited to inform herbivores, altering their foraging decisions and nudging them away from valued plants. Deploying informative odours in place of real plants sends a deceptive message and hence is a form of olfactory misinformation[14,15].

Our aim (Fig. 1) was to test the use of artificial informative odours, acting as virtual neighbours in a patch, to degrade perceived patch quality and alter herbivore foraging decisions. Specifically, we tested whether informative VOCs of an unpalatable (low-quality) plant species could replace real plants yet still provide associational refuge to a palatable seedling of another species. We recently presented a novel practical approach to find and quantify informative VOCs based on two main criteria of reliability, consistency and precision[16]. In this Article, we use and test this approach for its effectiveness in defining informative VOCs.

As our model for the study, we used a free-ranging macropod, the swamp wallaby (*Wallabia bicolor*), foraging in eucalypt woodland in eastern Australia. Swamp wallabies are native, abundant, mid-sized

[1]School of Life and Environmental Sciences, University of Sydney, Sydney, NSW, Australia. [2]Mammal Research Institute, Department of Zoology and Entomology, University of Pretoria, Pretoria, South Africa. ✉e-mail: patrick.finnerty@sydney.edu.au; clare.mcarthur@sydney.edu.au

**Fig. 1 | Conceptual model using patch-scale informative odour to protect palatable plants. a**, A palatable plant emitting odour providing a cue to foraging herbivores. **b**, A real avoided, low-quality plant neighbourhood provides associational refuge, protecting the palatable plant from herbivores by degrading actual patch quality and delaying browsing at a patch. **c**, A virtual neighbourhood of artificial informative odours mimicking real avoided neighbours replaces real plants yet still protects the palatable plant via associational refuge. **d**, Thus, population-level survival of **a** palatable plants is improved by **b** and **c** because many mammalian herbivores detect, identify and decide whether to visit and browse at food patches using odour.

(13–17 kg) browser/mixed feeders, ecologically equivalent to many species of deer, antelope and elephant in Europe, North America, Asia and Africa. Like these species, wallabies shape vegetative communities via selective browsing and are a known limiting factor to the recruitment and survival of plant species in regenerating or post-fire recovery areas[17,18]. Our unpalatable plant species was the pungent shrub, *Boronia pinnata*, and our palatable plant species was *Eucalyptus punctata*, a foundational canopy species. Both were part of the native vegetation open forest community.

To define the odour profile and determine putative informative VOC combinations for *B. pinnata*, we first collected 'headspace' VOC emissions from 30 plants across two sampling bouts (Extended Data Fig. 1 and Supplementary Note 1). We analysed these emissions using gas chromatography–mass spectrometry (GC-MS), producing the complete odour profile of 482 VOCs (see Supplementary data 1 and Extended Data Fig. 2 for a comparison of the odour profiles between sampling bouts). As many plant species emit VOCs in common, such VOCs may only convey information about a particular plant species if produced consistently and in distinct combinations for that species[16], usually described in terms of VOC pairs[19]. To identify putatively informative VOC pairs, we used two 'rules of reliability'[16]: VOC pairs needed to be emitted (a) consistently (by more than 50% of plants sampled) and (b) in precise proportions (between 0.5 (moderate precision) to 0 (absolute precision)). From these rules, we selected seven VOCs from a band of putatively informative VOC pairs (Extended Data Fig. 3): thujone, γ-terpinene, toluene, α-pinene, terpinolene, acetone and α-terpineol.

We then created three artificial odour treatments to act as virtual neighbours: informative, uninformative and flipped proportion. The informative treatment combined the seven VOCs (in six pairs) in correct informative proportions. The uninformative treatment combined seven new VOCs (in six pairs) that were recorded in *B. pinnata* but fell below our chosen reliability threshold. The flipped proportion treatment inverted the relative amount of informative VOCs within pairs (Extended Data Table 1). This treatment allowed us to test our prediction that the relative amounts of informative VOCs mattered, not simply their presence.

To test how swamp wallabies responded to the three virtual neighbour treatments compared to real *B. pinnata*, we deployed them in the field with three additional treatments: real *B. pinnata*, a procedural control and an untreated control. The real *B. pinnata* treatment was a single *E. punctata* seedling surrounded by five evenly spaced *B. pinnata* plants, allowing a direct comparison between real and virtual neighbours. The untreated control treatment was a single *E. punctata* seedling. The procedural control treatment was a single *E. punctata* seedling surrounded by five empty virtual neighbour odour dispensers (Extended Data Figs. 4 and 5) to ensure that any wallaby browsing effects were not due to the presence of the dispensers themselves.

All six treatments were deployed at our study site in plots ($n = 15$ per treatment, at least 50 m apart) in a completely randomized plot design. At each plot, five virtual or real neighbours were placed in a circle (radius 1 m) around a single *E. punctata* seedling at the centre of the patch (Extended Data Fig. 5). For the virtual neighbours, artificial odour (made up to a total of 2.96 ml) was deployed in a glass amber diffusion vial (design based on ref. 20; Extended Data Fig. 6) placed in a custom-built odour dispenser (to protect them from rain (Extended Data Fig. 4)). We had confirmed that the VOC emission rate from 2.96 ml for the informative treatment was equivalent to the VOC emission rate of a single real *B. pinnata* shrub. We had also confirmed that the proportional change in emissions from the virtual neighbour treatments stayed constant over time for at least 60 days, indicating a steady emission rate throughout the experiment. To assess the effectiveness of the treatments, we quantified time taken for a wallaby to first browse the *E. punctata* seedling in the plot.

Time to first wallaby browse on *E. punctata* seedlings differed significantly as a function of treatment (analysis of deviance likelihood ratio (LR) $\chi^2_5 = 74.70$, $P < 0.0001$; Fig. 2 and Extended Data Table 2). Informative virtual neighbours provided real browsing refuge from swamp wallabies, equivalent to protection provided by real *B. pinnata* plants ($P = 0.72$). Cox proportional hazard ratios indicated *E. punctata* seedlings were 20 times more likely to be browsed in the untreated control 'no neighbours' treatment than if surrounded by an informative virtual neighbourhood ($P < 0.0001$) and 17 times more likely to be browsed than if surrounded by a real neighbourhood of *B. pinnata* ($P < 0.0001$). *E. punctata* seedlings surrounded by flipped proportion and uninformative virtual neighbourhoods, as well as empty virtual neighbour vials (procedural control treatment), were equally likely to be browsed by wallabies as an *E. punctata* seedling 'alone' (Extended Data Table 2) (that is, no refuge effect). If *E. punctata* seedlings were browsed, wallabies generally ate all the foliage (75 of 82 seedlings) or most foliage (7 of 82 seedlings). Background wallaby activity did not differ across treatments (LR $\chi^2_5 = 1.90$, $P = 0.86$).

Our results show that informative virtual neighbourhoods provided associational browsing refuge to palatable seedlings, successfully replacing real neighbouring plants. We provide clear evidence that the specific subset of VOCs deployed in particular proportions designed to be informative were actually informative to wallabies and can be successfully deployed as olfactory misinformation to influence their foraging behaviour. Our results are a positive test for our approach[16] to find and quantify informative VOCs within the complex odour profile of an individual plant species. That the flipped proportion treatment did not provide refuge confirms that the relative proportion of the informative VOCs matter, not just their presence.

Our findings provide an important step forward in improving our understanding of both fundamental and applied mammalian behavioural ecology, providing new insight into the ways in which mammalian herbivores detect and respond to the world around them.

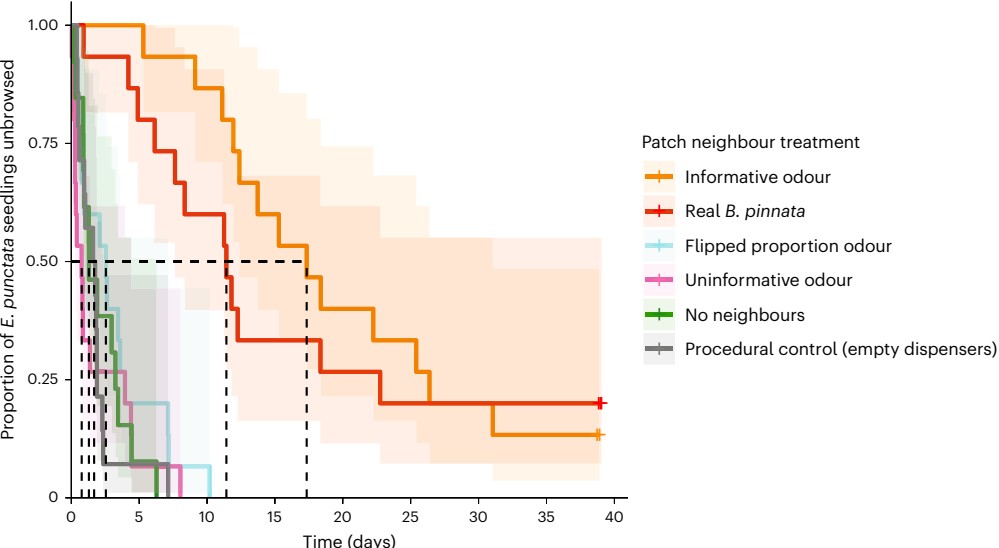

**Fig. 2 | Survival curve (time to first browse) of palatable *E. punctata* seedlings as a function of patch neighbour treatment (*n* = 15 per treatment).** The proportion of plots (±95% confidence intervals in shaded areas) remaining unbrowsed over 40 days. Cox proportional-hazard modelling showed a significant treatment effect (LR $\chi_5^2$ = 74.70, *P* < 0.0001). Dashed black lines indicate median survival time for each treatment.

We argue that our approach to detect and quantify informative VOCs has the potential to be applied more broadly to develop targeted virtual plant neighbours specific to herbivores in other systems.

As a management tool to protect palatable seedlings, virtual neighbours offer many advantages over real plants. Real plants compete for water and resources, which can outweigh protective effects in providing browsing refuge[21]. With future development, virtual neighbours could also be deployed en masse, quickly and likely cheaply, last long term or be removed at will, and tweaked over time to avoid potential habituation.

Herbivore browsing damage varies in detail and context globally: different plants, different herbivores, different landscapes. However, irrespective of the context, the logical approach used to define the putatively informative compounds of plant species is likely transferable to many mammalian (or potentially invertebrate) herbivores that rely primarily on plant odour information to forage. Consequently, using similar olfactory misinformation tactics, virtual neighbourhoods represent a new approach that has the potential to be harnessed as a benign, non-lethal, cost-effective and novel tool for reducing problem herbivory in conservation (for example, threatened plant species) and management (for example, forestry and agriculture) globally.

## Methods

### Animal ethics statement
Animal ethics approval was granted by the University of Sydney's Animal Ethics Committee (protocol number 2022/2196).

### Creating and deploying virtual neighbours
**Odour profile of *B. pinnata*.** To develop a complete odour profile of *B. pinnata*, we undertook 'headspace' VOC odour sampling of 30 naturally occurring *B. pinnata* plants across our study site in Ku-ring-gai Chase National Park, Sydney, Australia (33° 41′ 33″ S, 151° 08′ 44″ E) (Extended Data Fig. 1). Randomly selected individual plants sampled were of approximate equal height (198 ± 11 cm) and were at least 50 m away from any other sampled individual. Sampling was undertaken across two sample bouts (March 2021, *n* = 10, and April 2022, *n* = 20). Across both bouts, sampling was conducted between 9 a.m. and 4 p.m. Ambient temperatures recorded were similar across both bouts (March 2021, 20.8 °C to 24.3 °C; April 2022, 19.5 °C to 23.4 °C). Average daily rainfall was slightly higher in March 2021 (14.3 ± 4.7 mm) than in April 2022 (7.8 ± 3.0 mm).

To sample *B. pinnata* odour headspace, a polyacetate oven bag (Glad 35 cm × 48 cm) was placed over a single branch (branches used between individuals sampled were of approximate equal size, 6 × 'biounits' of 14 cm plant 'lengths'). Headspace was allowed to accumulate in the bag for 15 min. Next, air was extracted from the bag for 15 min through a thermal desorption tube filled with 200 mg each of Tenax TA (Markes International) using a PAS500 Personal Air Sampler (Spectrex). All thermal desorption tubes were analysed within 2 weeks of sampling by desorbing samples with automated thermal desorption (ULTRA 2 and UNITY 2, Markes International) for 6 min at 300 °C and concentrated on a Tenax TA cold trap at −30 °C. This cold trap was then flash heated to 300 °C, and the concentrated sample injected splitless via a heated transfer line (150 °C) onto a 7890A GC-MS (Agilent Technologies) fitted with a BP1 capillary column (60 m × 0.32 mm, 1 μm film thickness; Agilent) at a flow rate of 2.3 ml min$^{-1}$. The GC oven was heated at 35 °C for 5 min then 4 °C min$^{-1}$ to 160 °C then 20 °C min$^{-1}$ to 300 °C for 10 min. The GC was coupled to a mass-selective detector (Model 5975C; Agilent). The temperature of the GC-MS interface was 280 °C, the MS ion source 230 °C and the quadrupole 150 °C. The detector, in electron impact mode (70 eV), scanned the range of 35–300 *m/z*. Operation of the GC-MS was controlled by Agilent Chemstation (version E.02.01.117) and the ULTRA 2 and UNITY 2 by Maverick (version 5.0; Markes).

Common contaminating ions (73, 84, 147, 149, 207, 221 and 281 *m/z*) were removed from the chromatograms using the Denoising function in OpenChrom (version 1.1.0 (ref. 22)). Chromatograms were then processed using the MSeasyTkGUI package[23] in RStudio (version 4.2.0, R Development Core Team 2022) and all putative compounds clustered. MSeasyTkGUI also produced peak areas for the putative compounds based on their total ion count (TIC). Blank samples (*n* = 7) were run in conjunction with all analysis; the upper 95% confidence interval of the mean blank value was subtracted from all samples. Final ion counts of the putative compounds emitted by our plants were obtained by subtraction of the background TIC recorded for each compound from the plant samples. Identification of putative compounds was made by a combination of manually comparing mass spectra against a commercial library (NIST14 library in NIST MS Search v.2.2f; NIST) and the library's calculated match factor, using a threshold of 700. The final

TIC of those putative compounds identified as the same compound was added up to obtain only one value per compound. In total, we identified 482 individual VOCs across all *B. pinnata* sampled from after blank subtraction (see Supplementary Note 1, Supplementary Data 1 and Extended Data Fig. 2 for a comparison of the odour profiles between sampling bouts).

**Defining the informative VOCs of *B. pinnata* and developing virtual neighbour treatments.** After using the two 'rules of reliability'[16], we selected seven VOCs from a band of putatively informative VOC pairs for *B. pinnata* (Extended Data Fig. 3) and combined them in appropriate proportions (based on average TICs for each VOC, from A1 above) to create (a) informative virtual neighbour treatment. To create (b) uninformative virtual neighbour treatment, we combined seven new VOCs and pairs that fell below our chosen reliability threshold: *d*-limonene, (1*R*)-(−)-myrtenal, nonanoic acid, β-myrcene, tridecane, 3-pentanone and geraniol, and to create (c) flipped proportion virtual neighbour treatment, we inverted the relative amount of informative VOCs within pairs (Extended Data Table 1).

We combined VOCs in the identified paired proportions so that the total volume of the mixture was 1 ml per treatment. Mixtures were measured into 10 ml glass amber virtual neighbour vials (Agilent) sealed with polytetrafluoroethylene-lined butyl septa headspace caps (Agilent). Vials were pierced with a diffusion tube made from a 20-gauge syringe needle (Sigma-Aldrich) attached to a polypropylene solid-phase extraction tube containing a 20 µm porosity polyethylene frit to limit the diffusion rate (based on ref. 24; Extended Data Fig. 6).

As the GC-MS has different sensitivities to different compounds, we verified whether compounds mixed in virtual neighbour vials for each treatment matched the paired proportions from the *B. pinnata* plants. VOC samples from virtual neighbour vials (*n* = 10 per treatment) were measured dynamically using 1 l glass Mason jars (Ballmason Australia) where the jar lid was fitted with 1/4 inch brass bulkhead fittings (Swagelok) to allow air in and out of the jar (Extended Data Fig. 7). Instrument air (BOC), passed through an activated charcoal scrubber, was supplied to the Mason jar at 1.3 l min$^{-1}$ using a mass flow controller (Aalborg). VOCs were collected from the jar outlet using a sorbent tube containing 200 mg Tenax TA (Markes International) connected to an air pump (AirChek 2000; SKC) flowing air at 70 ml min$^{-1}$ for 15 min. Vials were allowed 15 min to acclimate to conditions within the jar before VOC sampling commenced. Background (control) samples were taken at the beginning and end of each day. Post-sampling tubes were maintained at 4 °C until analysis by GC-MS. Compounds were identified after GC-MS analysis using the same protocol used when measuring the odour profiles from *B. pinnata* plants. Emission rates (µg h$^{-1}$) of each of the compounds were determined using their background subtracted concentrations, the chamber flow rate and the sampling duration. We adjusted compound volumes in the virtual neighbour treatments so their emissions matched those from the real plants.

**Comparing emission rates of virtual neighbours to real *B. pinnata* neighbours.** In our manipulative experiment, we deployed *B. pinnata* plants sourced from a nursery (Plants Plus Cumberland Forest Nursery, West Pennant Hills, Sydney) as neighbours. Therefore, we next compared and again adjusted our virtual neighbour odours so the relative proportions of VOC emissions and the absolute emission rates matched those of these plants (*n* = 8; height 650 ± 12 mm, biomass 45.7 ± 2.5 g above ground dry weight, calculated after odour sampling).

VOC samples were taken from a single *B. pinnata* branch (still attached to the main plant) inserted inside of a custom-built, 9 l branch enclosure (Extended Data Fig. 8). The two ends of the chamber were made from polytetrafluoroethylene supporting a transparent enclosure made from polyvinyl fluoride film (Dupont Chemicals). Ambient air, passed through an activated charcoal scrubber, was supplied to the chamber at 12 l min$^{-1}$ using a mass flow controller

(Aalborg). Supplementary photosynthetically active radiation (PAR) (380 µmol m$^{-2}$ s$^{-1}$) was provided by 20 W LED lights (Arlec). Mean air temperature inside the chamber was 23.3 °C. PAR and air temperatures inside the chamber were recorded automatically every minute using a Hobo H21 Micro Station Datalogger coupled with SLIAM003 PAR and S-THB-M002 temperature/relative humidity sensors (Onset). VOCs were collected from the enclosures using a sorbent tube containing 200 mg Tenax TA (Markes International) connected to an air pump (AirChek 2000; SKC) flowing air at 200 ml min$^{-1}$ for 30 min. *B. pinnata* branches were allowed 15 min to acclimate to conditions within the enclosure before VOC sampling commenced. Background (control) enclosure samples were taken at the beginning and end of each day. Post-sampling tubes were maintained at 4 °C until analysis by GC-MS. Compounds were identified after GC-MS analysis using the same protocol used when developing *B. pinnata* odour profile.

Quantification of the compounds was made using the three major characteristic ions of the compounds in comparison to external standards diluted in methanol. All chemicals were purchased from Sigma-Aldrich. Emission rates (mg g(dw)$^{-1}$ h$^{-1}$) of each of the compounds were determined using their background subtracted concentrations, the chamber flow rate, sampling duration and the dry weight (dw) of the leaves of each branch. The mean emission rate per compound across all plants was summed and multiplied by the mean branch dry weight to determine the mean whole plant emission rate (mg h$^{-1}$). Comparison with the vial emission rate showed that the whole plant emissions were on average 2.96 times greater than 1 ml vials. Hence, the volume of the compound mixtures was increased to 2.96 ml to give a comparable emission rate to the plants (Extended Data Table 1). Emissions of the informative VOCs from nursery plants were similar to those of the wild plants (Extended Data Fig. 2).

**Virtual neighbour emission rate over time.** Ten replicates of informative, flipped proportion and uninformative virtual neighbour vials were created to a volume of 2.96 ml, and the total weight of each vial was measured. All vials were placed on a heated plate under laboratory conditions (DBH20D dry block heater, Ratek Instruments) set at a constant 25 °C. Weights of vials were measured every 7 days and the slope of the weight loss over time determined to give an average emission rate (mg h$^{-1}$).

**Virtual neighbour odour dispenser.** To deploy virtual neighbour vials at our study site, ensuring vials were secure and were not affected by rain, we created bespoke odour dispensers (Extended Data Fig. 4). Dispensers did not alter the VOC emission from virtual neighbours (analysis of similarity indicated no significant difference in VOCs emitted from virtual neighbour vials alone or when placed in dispensers, *R* = 0.092, *P* = 0.13, *n* = 10). To account for dispenser presence affecting wallaby foraging behaviour, we included a fourth 'procedural control' treatment comprising an *E. punctata* seedling surrounded by five evenly spaced dispensers with empty virtual neighbour vials.

**Associational refuge main trial.** All six treatments were deployed at our study site in plots (*n* = 15 per treatment, at least 50 m apart) in a completely randomized plot design. At each plot, five virtual or real neighbours were placed evenly in a circle (radius 1 m) around a single *E. punctata* seedling (325 ± 18 mm tall; Extended Data Fig. 5). All plants were sourced from Plants Plus Cumberland Forest Nursery and came potted (using Scott's Osmocote Native Premium Potting Mix) in black plastic 200 mm 'Garden City Plastic Grow Plant Pots'. Temperature ranged across the study period from 13.3 °C to 37.0 °C with a mean of 6.5 mm daily rainfall, with 12 days of rain (of >1 mm) over the total 40 day period (Terrey Hills, Sydney, Bureau of Meteorology 2022).

Patches were monitored for 40 days between February and March 2023 using motion-triggered infra-red trial cameras (ScoutGuard SG560K or SG2060-K; Professional Trapping Supplies). Cameras were

fastened to wooden stakes (camera height = 0.7 m, distance to seedling = 1.5 m) at an approximate 45° angle towards the palatable seedlings. Cameras were set to record 60 s videos with instant re-trigger.

After 40 days, we quantified the survival time of palatable seedlings at 'time to first wallaby browse (days)' (when a wallaby consumed any part of the palatable seedling). If browsed, we estimated the percentage of foliage consumed from each of *E. punctata* seedlings as seen on camera using a visual estimate, with percentage intervals of 0%, 25%, 50%, 75% and 100% eaten.

**Pre-trial period.** Before the main trial, we ran a 14 day pre-trial period to both habituate wallabies to the experimental set-up of camera and stake and calculate a score of background wallaby activity per patch (background wallaby activity score = the number of wallabies recorded at a treatment site during the pre-trial period).

**Statistical analysis.** We used generalized linear models with a Poisson distribution and log link function in R (version 4.2.0; R Core Team, 2022, lme4 package[25]) to test whether there was a difference in background wallaby activity (dependent variable) between treatment sites.

We used Cox proportional-hazards models in R ('survival' package[26]) to model 'survival' (where failure is based on time to first browse) as a function of treatment and background wallaby activity (fixed factors). These models were also used to calculate pairwise hazard ratios between treatments. These analyses take into account right-censored data. Data were censored for any un-browsed seedlings by using the maximum number of hours until the end of the experimental period. We report the hazard ratio (exp(coef)) for all pairwise comparisons between treatments (Extended Data Table 2).

When significant differences existed, we performed Tukey post hoc tests to locate those differences (reported using alphabetical superscript in figures).

### Reporting summary

Further information on research design is available in the Nature Portfolio Reporting Summary linked to this article.

## Data availability

The datasets generated during and/or analysed during the current study are available in the Sydney eScholarship Repository[27] (https://hdl.handle.net/2123/31657). Supplementary Data 1 provides a complete odour profile from odour headspace sampling undertaken.

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

## Acknowledgements

Figure 1 was prepared by J. Bytheway. P.B.F. acknowledges funding from the Ecological Society of Australia (Jill Landsberg Trust Fund Scholarship, grant number JLTF A10656310), the Australian Academy of Science (Max Day Environmental Science Fellowship Award, grant number MDF230142803), the University of Sydney and New South Wales Department of Planning and Industry Memorandum of Understanding Partner Grant (grant number DOC21/445547), the Royal Zoological Society of Australia (Paddy Pallin Science Grant, grant number RZNSWPP2021, and Ethel Mary Read (EMR) Research Grant, grant number RZNSWEMR2021) and the Australian Wildlife Society (Student Research Grant, grant number AWSSRG2021PF). C.M., P.B.B., M.P. and C.J.P. acknowledge funding from the Australian Research Council (grant number DP190101441).

## Author contributions

P.B.F., C.M., P.B.B., C.J.P. and A.M.S. conceived the ideas and designed the methodology; P.B.F. and M.P. collected the data; P.B.F., M.P., C.G.O. and C.M. analysed and interpreted the data; all authors contributed critically to the manuscript drafts and gave final approval for publication.

## Funding

## Competing interests

The authors declare no competing interests.

## Additional information

**Extended data** is available for this paper at https://doi.org/10.1038/s41559-024-02330-x.

**Correspondence and requests for materials** should be addressed to Patrick B. Finnerty or Clare McArthur.

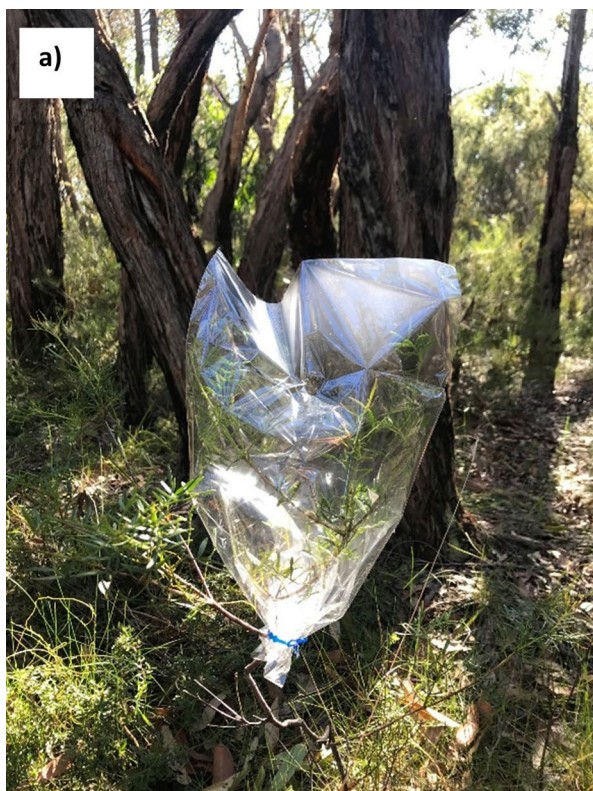

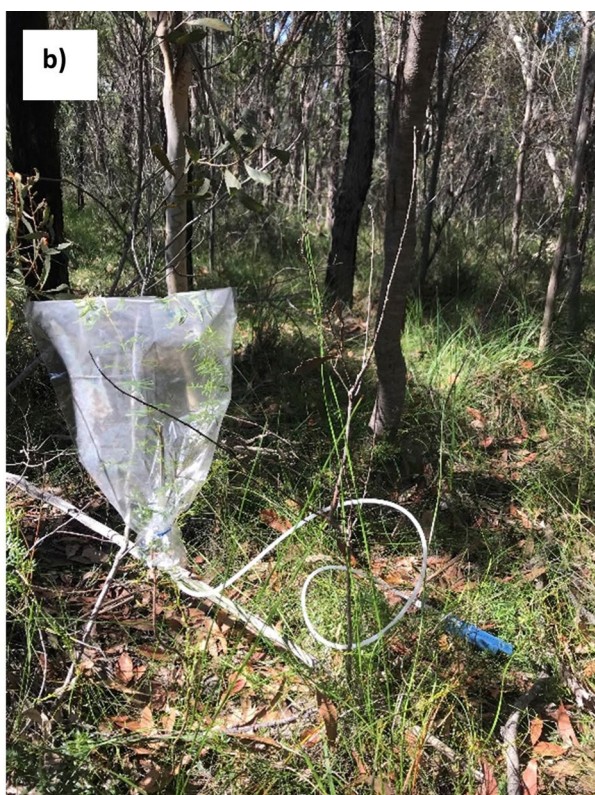

**Extended Data Fig. 1 | Odour 'headspace' sampling of B. pinnata.** Odour 'headspace' sampling of B. pinnata for GC-MA analysis. **a)** Polyacetate oven bag over a single B. pinnata branch. Odour headspace was allowed to accumulate in the bag for 15 minutes prior to sampling. **b)** Air was extracted from the bag for 15 minutes through a thermal desorption tube filled with 200 mg of Tenax TA using a PAS500 Personal Air Sampler.

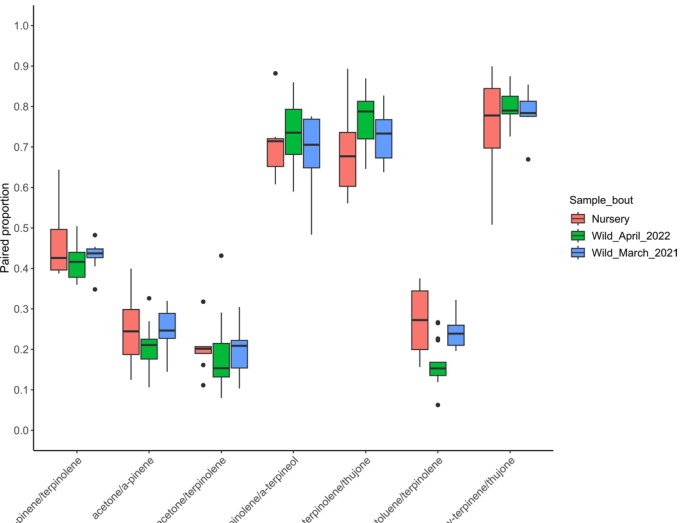

**Extended Data Fig. 2 | Paired VOC proportions across *B. pinnata* odour sampling bouts.** Boxplots of paired VOC proportions sampled for all seven selected informative VOC pairs, across two sampling bouts for odour headspace of wild B. pinnata plants (March 2021, n = 10, April 2022, n = 20) and from branch enclosure sampling of nursery raised B. pinnata (n = 8). Plots represent the proportion of the first listed compound in each VOC pair (calculated after fourth-root transformation, see Methods section). Boxplots indicate the median, the first and third quartiles, and the maximum and minimum values. Closed circles indicate outliers.

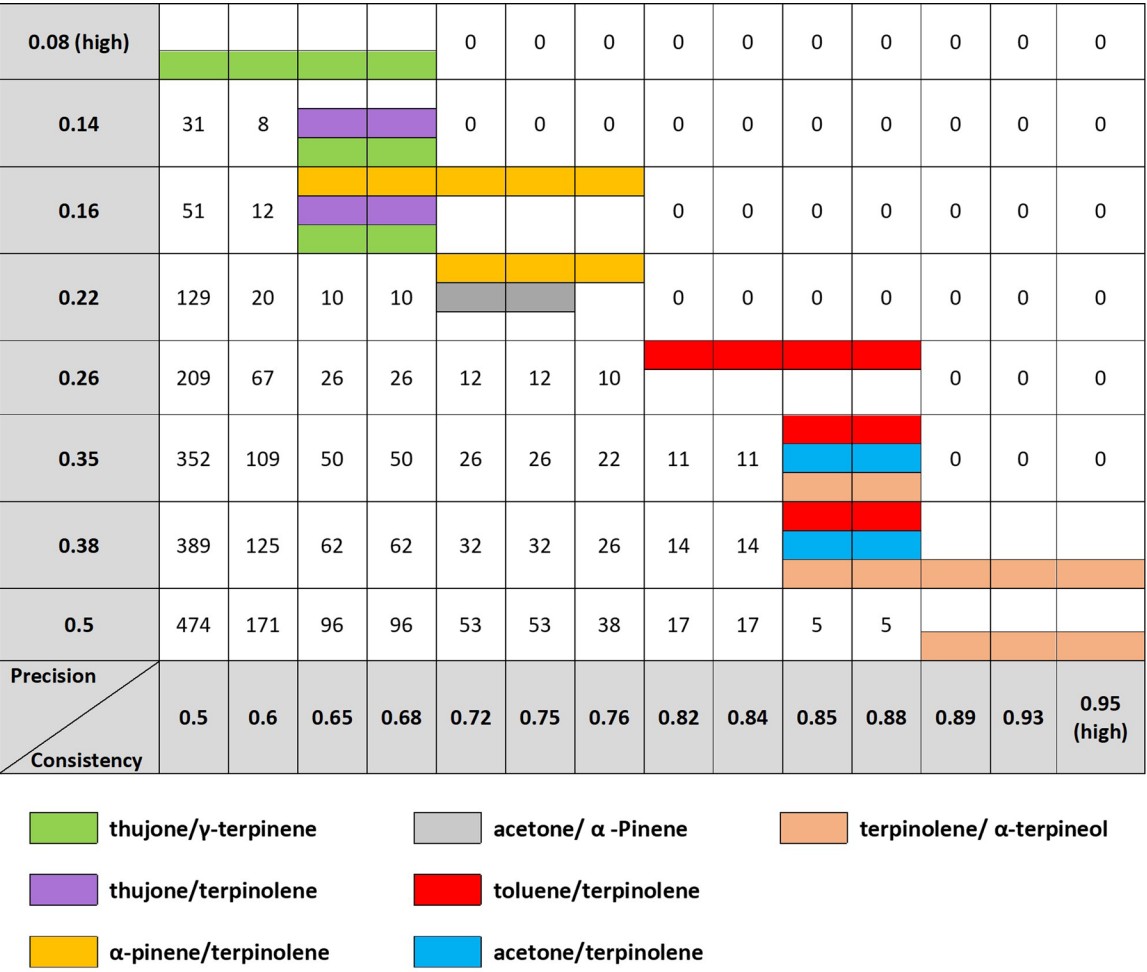

**Extended Data Fig. 3 | Informative *B. pinnata* VOC consistency-precision reliability space.** Informative pairs of volatile organic compounds (VOCs) identified from B. pinnata presented in a consistency-precision reliability space. Here, values for the two 'Rules of reliability' are present on a scale of consistency (threshold baseline 0.5 to high consistency 0.95) and precision (threshold baseline 0.5 to high precision 0.08). Specific VOC pairs, presented in a unique colour, are shown when one to three pairs were detected. Where a cell contains the number 0, no pairs were identified. Where more than three pairs were identified, the number of pairs is shown.

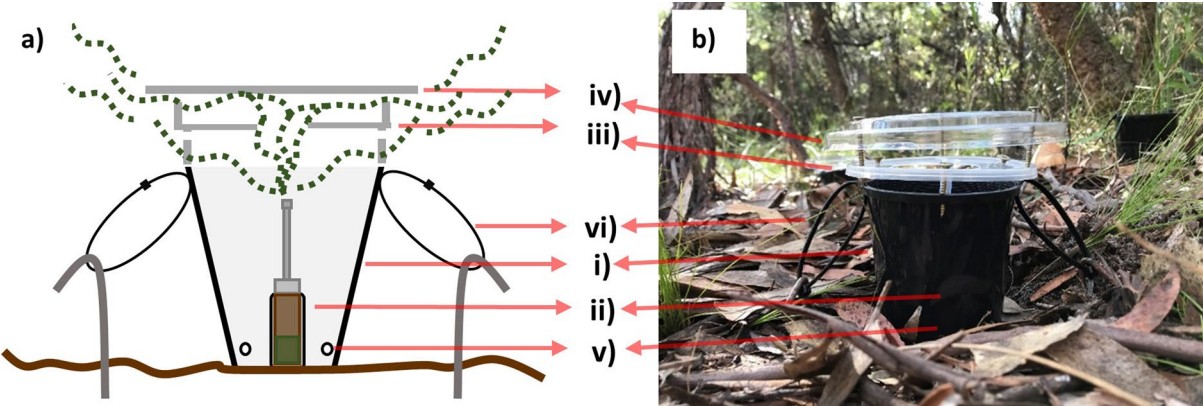

**Extended Data Fig. 4 | Virtual neighbour odour dispenser. a)** A longitudinal schematic of a virtual neighbour odour dispenser (i) a black plastic 200 mm 'Garden City Plastic Grow Plant Pot' with a (ii) 2 cm piece of 19 mm black poly pipe inserted inside to securely fit a single vial. 5 mm above the open pot, using screws we attached a (iii) 220 mm round clear plastic lid with a 50 mm hole cut centrally above the vial to allow the odour from vials to escape. We placed (iv) an intact, larger 300 mm round clear plastic lid 5 mm above the first lid as a secondary rain guard. (v) Three small holes were drilled into the base of the pots to act as a flue to produce an air draft to help disperse odour from vials. (vi) Two 300 × 4.8 mm black cable ties were attached to the sides of each pot, which acted as an anchor point for tent pegs which were used to secure dispensers to the ground. Odour being emitted from vials is depicted as dashed green lines. **b)** A virtual neighbour odour dispenser attached to the ground in-situ at our study site.

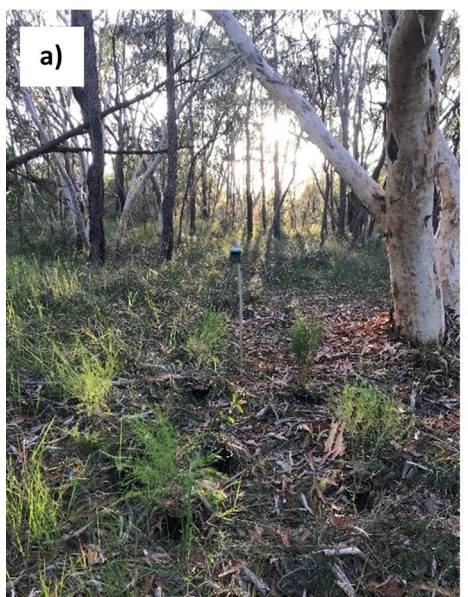

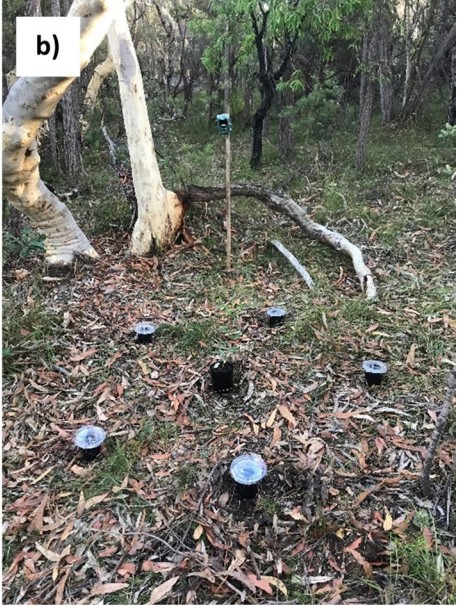

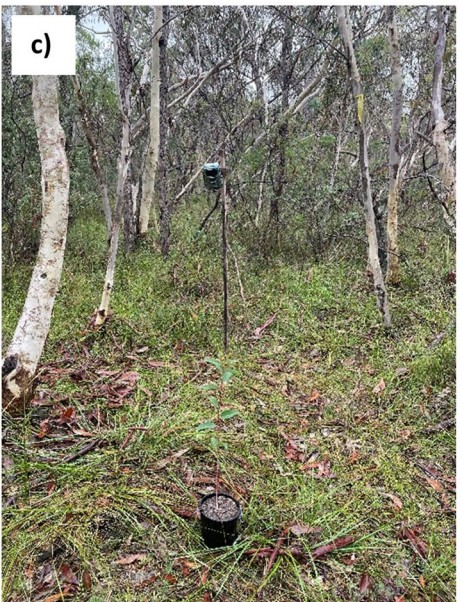

**Extended Data Fig. 5 | Associational refuge main trial treatments.**
Associational refuge main trial treatments **a)** real neighbourhood treatment of a single E. punctata surrounded by five B. pinnata, **b)** virtual neighbourhood manipulation treatment of a single E. punctata surrounded by five virtual neighbour odour dispensers, **c)** untreated control, a single E. punctata on its own, against background vegetation. Images show experimental set up of motioned-triggered cameras fastened to wooden stakes (camera height = 0.7 m, distance to seedling = 1.5 m) at an approximate 45° angel towards the focal seedlings.

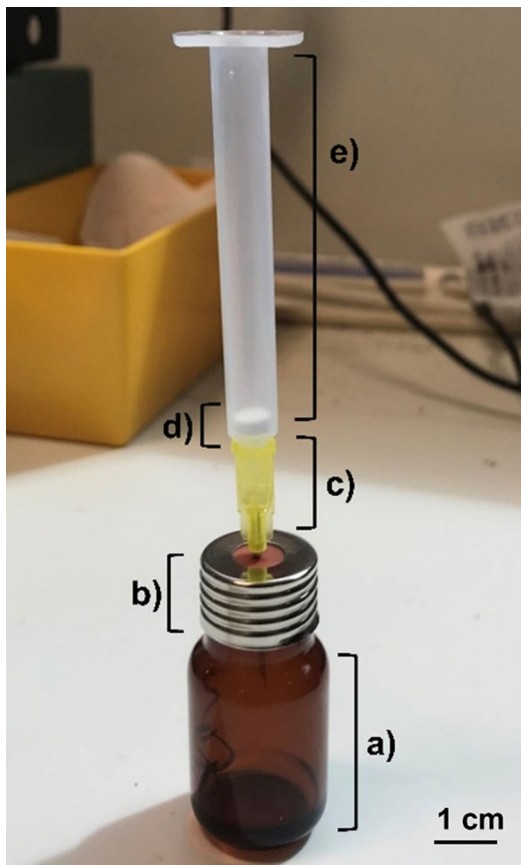

**Extended Data Fig. 6 | A virtual neighbour vial.** A virtual neighbour vial comprising 2.96 ml VOC compound mixture in a (a) 10 mL glass amber vial, sealed with (b) PTFE lined butyl septa headspace cap. Vials are pierced with a diffusion tube made from a (c) 20 gauge syringe needle attached to a (e) polypropylene SPE tube containing a (d) 20μ porosity PE frit to limit the diffusion rate.

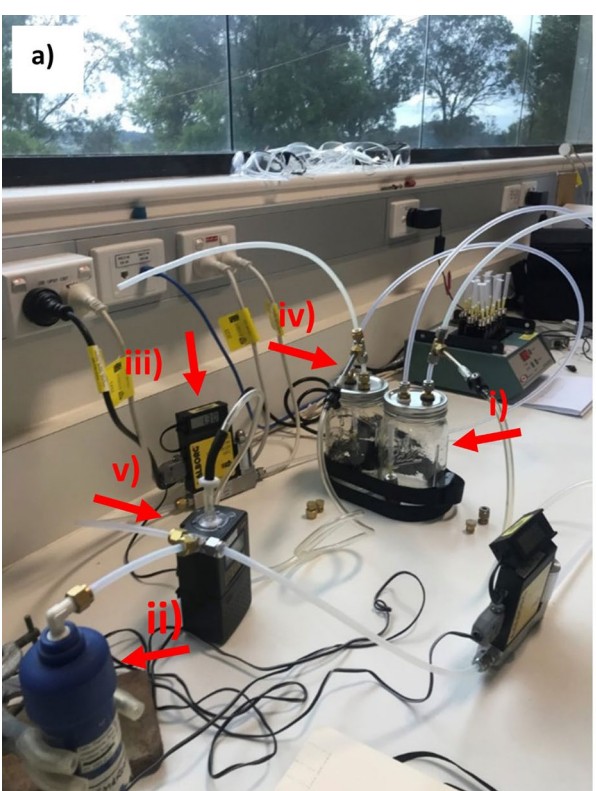
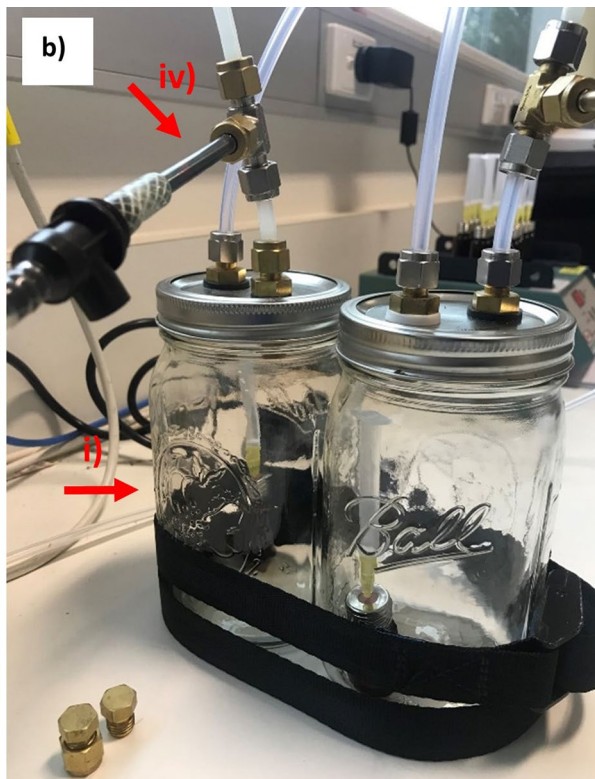

**Extended Data Fig. 7 | Virtual neighbour VOC sampling.** a) Overview of virtual neighbour VOC sampling set-up. b) Virtual neighbours in mason jars undergoing VOC sampling. Virtual neighbour VOC sampling included (i) one litre glass Mason jars with lids fitted with ¼" brass bulkhead fittings to allow air in and out of the jar. Instrument air, passed through an activated charcoal scrubber (ii), was supplied to the Mason jar at 1.3 L min-1 using a mass flow controller (iii). VOCs were collected from the jar outlet using a sorbent tube containing 200 mg Tenax TA (iv) connected to an air pump (v) flowing air at 70 mL min-1 for 15 minutes.

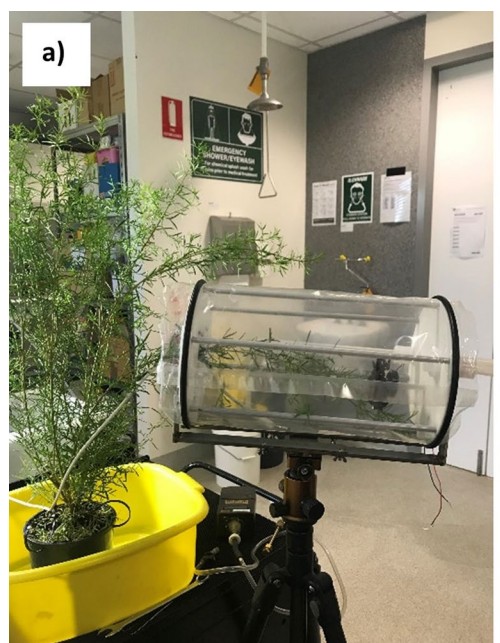
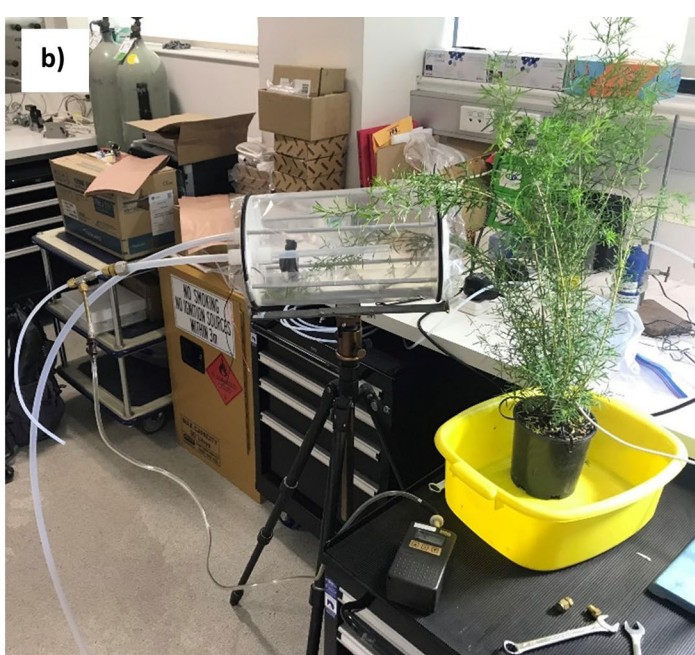

**Extended Data Fig. 8 | Sampling B. pinnata VOC emission rate.** a) Here, a single branch of a B. pinnata was paced inside a custom-built branch enclosure chamber. b) An air pump, connected to the chamber via plastic tubing draws air out of the chamber, and through a sorbent tube where the VOCs emitted by the branch are collected. Later, sorbent tubes were run through GC-MS to determine VOCs emitted and to calculate 'branch' emission rate. Total above ground plant dry weight (g) was divided by branch dry weight and then multiplied by branch emission rate to give total plant emission rate (mg VOC / hour).

**Extended Data Table 1 | Compound volumes used in virtual neighbour treatments**

| | *thujone* | *γ-terpinene* | *terpinolene* | *α-pinene* | *acetone* | *toluene* | *α-terpineol* |
|---|---|---|---|---|---|---|---|
| **Informative** | 482.1 µL | 755.0 µL | 1445.0 µL | 93.7 µL | 0.1 µL | 0.4 µL | 184.0 µL |
| **Flipped proportion** | 5.3 µL | 0.3 µL | 0.1 µL | 0.4 µL | 2461.6 µL | 218.1 µL | 274.2 µL |
| | *D-limonene* | *(1R)-(-)-myrtenal* | *nonanoic acid* | *ß-myrcene* | *tridecane* | *3-pentanone* | *geraniol* |
| **Uninformative** | 187.7 µL | 1.0 µL | 1.1 µL | 74.2 µL | 2.1 µL | 0.2 µL | 2693.6 µL |

Compound volumes used in virtual neighbour treatments, adjusted for exact compound concentration and volume to match real B. pinnata VOC paired proportions and emission rate. All virtual neighbour vials were mixed to a volume of 2.96 mL.

**Extended Data Table 2 | Virtual neighbourhood time to first wallaby browsed**

**2i)**

| Pairwise comparison | Hazard ratio | p-value |
|---|---|---|
| Untreated control (no neighbours) vs Informative odour | **19.79** | **< 0.0001** |
| Untreated Control (no neighbours) vs Real *B. pinnata* | **17.10** | **< 0.0001** |
| Real *B. pinnata* vs Informative odour | 1.16 | 0.72 |
| Flipped proportion odour vs Informative odour | **12.39** | **< 0.0001** |
| Uninformative odour vs Informative odour | **22.95** | **< 0.0001** |
| Procedural control (empty dispenser) vs Informative odour | **22.26** | **< 0.0001** |
| Flipped proportion odour vs Real *B. pinnata* | **10.71** | **< 0.0001** |
| Uninformative odour vs Real *B. pinnata* | **19.84** | **< 0.0001** |
| Procedural control (empty dispenser) vs Real *B. pinnata* | **19.24** | **< 0.0001** |
| Uninformative odour vs Flipped proportion odour | 1.85 | 0.11 |
| Procedural control (empty dispenser) vs Flipped proportion odour | 1.80 | 0.15 |
| Untreated control (no neighbours) vs Flipped proportion odour | 1.60 | 0.27 |
| Procedural control (empty dispenser) vs Uninformative odour | 0.97 | 0.93 |
| Untreated control (no neighbours) vs Uninformative odour | 0.86 | 0.71 |
| Untreated control (no neighbours) vs Procedural control (empty dispenser) | 0.89 | 0.76 |
| **Hazard Ratio (HR) indicators:** | | |
| HR = 1: no effect | | |
| HR < 1 = reduction in hazard (for first listed treatment) | | |
| HR > 1 = increase in hazard (for first listed treatment) | | |

**2ii)**

| Treatment | Median (days) | S.E. | N. visits |
|---|---|---|---|
| Untreated control (no neighbours) | 1.3 | 0.50 | 13 |
| Informative odour | 14.51 | 2.06 | 13 |
| Real *B. pinnata* | 9.81 | 1.20 | 12 |
| Flipped proportion odour | 2.55 | 0.77 | 15 |
| Uninformative odour | 0.77 | 0.20 | 15 |
| Procedural control (empty dispensers) | 1.69 | 0.45 | 14 |

**2i)** Virtual neighbourhood time to first wallaby browsed hazard ratios for pairwise comparisons between individual treatments. Significant treatment effect (Cox proportional-hazards model, LR χ_5^2=74.70, P<0.0001). Hazard ratios indicate the likelihood that a treatment (first listed) will be browsed in comparison to another (second listed). High hazard ratios indicate a higher probability of being browsed. **2ii)** Of patches browsed, summary of median time to first wallaby browse in days, standard error (S.E) and number of browsing visits (N. visits).

# Reporting Summary

## Statistics

For all statistical analyses, confirm that the following items are present in the figure legend, table legend, main text, or Methods section.

| n/a | Confirmed | |
|---|---|---|
| ☐ | ☒ | The exact sample size (*n*) for each experimental group/condition, given as a discrete number and unit of measurement |
| ☐ | ☒ | A statement on whether measurements were taken from distinct samples or whether the same sample was measured repeatedly |
| ☐ | ☒ | The statistical test(s) used AND whether they are one- or two-sided<br>*Only common tests should be described solely by name; describe more complex techniques in the Methods section.* |
| ☐ | ☒ | A description of all covariates tested |
| ☐ | ☒ | A description of any assumptions or corrections, such as tests of normality and adjustment for multiple comparisons |
| ☐ | ☒ | A full description of the statistical parameters including central tendency (e.g. means) or other basic estimates (e.g. regression coefficient) AND variation (e.g. standard deviation) or associated estimates of uncertainty (e.g. confidence intervals) |
| ☐ | ☒ | For null hypothesis testing, the test statistic (e.g. $F$, $t$, $r$) with confidence intervals, effect sizes, degrees of freedom and $P$ value noted<br>*Give P values as exact values whenever suitable.* |
| ☒ | ☐ | For Bayesian analysis, information on the choice of priors and Markov chain Monte Carlo settings |
| ☒ | ☐ | For hierarchical and complex designs, identification of the appropriate level for tests and full reporting of outcomes |
| ☒ | ☐ | Estimates of effect sizes (e.g. Cohen's *d*, Pearson's *r*), indicating how they were calculated |

*Our web collection on statistics for biologists contains articles on many of the points above.*

## Software and code

Policy information about availability of computer code

| Data collection | No software was used |
|---|---|
| Data analysis | We analysed the data collected in this study using R version 4.2.0. |

For manuscripts utilizing custom algorithms or software that are central to the research but not yet described in published literature, software must be made available to editors and reviewers. We strongly encourage code deposition in a community repository (e.g. GitHub). See the Nature Portfolio guidelines for submitting code & software for further information.

## Data

Policy information about availability of data

All manuscripts must include a data availability statement. This statement should provide the following information, where applicable:

- Accession codes, unique identifiers, or web links for publicly available datasets
- A description of any restrictions on data availability
- For clinical datasets or third party data, please ensure that the statement adheres to our policy

The datasets generated during and/or analysed during the current study are available in the Sydney eScholarship Repository, https://hdl.handle.net/2123/31657

# Research involving human participants, their data, or biological material

Policy information about studies with <u>human participants or human data</u>. See also policy information about <u>sex, gender (identity/presentation), and sexual orientation</u> and <u>race, ethnicity and racism</u>.

| | |
|---|---|
| Reporting on sex and gender | Not applicable |
| Reporting on race, ethnicity, or other socially relevant groupings | Not applicable |
| Population characteristics | Not applicable |
| Recruitment | Not applicable |
| Ethics oversight | Not applicable |

Note that full information on the approval of the study protocol must also be provided in the manuscript.

# Field-specific reporting

Please select the one below that is the best fit for your research. If you are not sure, read the appropriate sections before making your selection.

☐ Life sciences    ☐ Behavioural & social sciences    ☒ Ecological, evolutionary & environmental sciences

For a reference copy of the document with all sections, see <u>nature.com/documents/nr-reporting-summary-flat.pdf</u>

# Ecological, evolutionary & environmental sciences study design

All studies must disclose on these points even when the disclosure is negative.

| | |
|---|---|
| Study description | Stage 1 involved replicating the informative odour compounds of an avoided plant species, Boronia pinnata, to develop virtual neighbours. We sampled the odour 'headspace' of multiple real B. pinnata at our study site (n = 30) to develop a complete odour profile for this species. We then employed two 'Rules of reliability' to define the informative volatile organic compounds (VOCs) of B. pinnata. Identified informative VOCs (seven VOCs, into six pairs) were then mixed together into glass amber diffusion vials to form our informative virtual neighbour treatment. As a comparison, we combined the same number of VOCs (seven VOCs, into six pairs) that were recorded in B. pinnata but fell below our chosen reliability threshold as an uninformative virtual neighbour treatment. We inverted the relative amount of informative VOCs within pairs as a third flipped proportion virtual neighbour treatment. We then analysed the VOC emissions from these virtual neighbour vials (n = 10 per treatment, 30 total) and adjusted VOC volumes to ensure that VOCs were being emitted in the correct paired proportions identified from B. pinnata. Finally, we compared the VOC emission rate between informative virtual neighbour vials (n = 8) and real B. pinnata (n = 8), and adjusted VOC volumes to ensure that VOC emission rates were equivalent.<br><br>Stage 2 involved testing how swamp wallabies, Wallabia bicolor, responded to the three virtual neighbour treatments compared to real B. pinnata. This study was conducted on free-ranging wallabies in eucalypt woodland in eastern Australia. Virtual neighbours and real B. pinnata were deployed at our study site in plots (n = 15 per treatment, at least 50 m apart) in a completely randomised plot design. At each plot, five virtual or real neighbours were placed in a circle (radius 1 m) around a single highly palatable Eucalyptus punctata seedling at the centre of the plot. Virtual neighbour vials were deployed in bespoke virtual neighbour odour dispensers. As part of this experiment we also compared two additional treatments: a procedural control, and an untreated control (n = 15 per treatment). The untreated control treatment was a single E. punctata seedling. The procedural control treatment was a single E. punctata seedling surrounded by five empty virtual neighbour odour dispensers to ensure that any wallaby browsing effects were not due to the presence of the dispensers themselves.<br><br>Plots were monitored for 40 days between February and March 2023 using motion-triggered infra-red trail cameras. After 40 days, we quantified the survival time of E. punctata seedlings at 'time to first wallaby browse (days)' (when a wallaby consumed any part of the palatable seedling). If browsed, we quantified the proportion of E. punctata biomass consumed after the first browsing visit. We also recorded whether the palatable seedling was browsed during the day or during the night. Before this experiment, we ran a 14-day pre-trial period to both habituate wallabies to the experimental set-up of camera and stake and calculate a score of background wallaby activity per plot. |
| Research sample | Free-ranging swamp wallaby (Wallabia bicolor) population within Ku-ring-gai Chase National Park, Sydney, Australia (33°41'33"S, 151°08'44"E) |
| Sampling strategy | No sample size calculation was performed prior to the experiment, as the underlying variation in the data was not known. Our sample size was chosen based on previous work done on a similar scale (Bedoya‑Pérez et al. 2014 Oecologia, Finnerty et al. 2017 J Anim Ecol, Orlando et al. 2020 Bio Letters) |
| Data collection | Stage 1: Data was collected both from the field, and in the lab by two observers. Stage 2: Data was collected from footage collected |

| | |
|---|---|
| Data collection | by motion-triggered infra-red trail cameras placed directly in the field by one observer. |
| Timing and spatial scale | Stage 1: B.pinnata odour sampling was undertaken across two sampling bouts (March 2021, n = 10 and April 2022, n = 20). Randomly selected individual plants sampled were of approximate equal height (198 ± 11 cm) and were at least 50 m away from any other sampled individual. Development of virtual neighbours was completed between April 2022 - February 2023.<br><br>Stage 2: Treatments were deployed and camera-trapping occurred between February 2023 - March 2023. Treatment plots and cameras were spaced at least 50 m apart. |
| Data exclusions | No data was excluded from our analysis. |
| Reproducibility | No measures have been taken to verify the reproducibility of the experimental findings. |
| Randomization | All six treatments were deployed at our study site in plots in a completely randomised plot design. |
| Blinding | Complete blinding was not possible as treatments could be partially determined from camera trap footage. However, as virtual neighbour vials were deployed in blackened odour dispensers, informative, uninformative, flipped proportion, and procedural control treatments could not be distinguished from one another. There was also little to no room for interpretation as data collection was a simple recording of when a wallaby first browsed on a seedling within a plot, how much of the seedling was consumed, and for how long. |

Did the study involve field work?  ☒ Yes  ☐ No

# Field work, collection and transport

| | |
|---|---|
| Field conditions | Stage 1: B. pinnata odour sampling at our study site was conducted cross two bouts in March 2021 and April 2022, between 8 am to 5 pm respectively. Ambient temperatures recorded were similar across both bouts (March 2021: 20.8°C – 24.3°C, April 2022: 19.5°C – 23.4°C) and average daily rainfall was slightly higher in March 2021 (14.3 ± 4.7 mm) than in April 2022 (7.8 ± 3.0 mm).<br><br>Stage 2: Between February and March 2023 temperature at our study site ranged from 13.3 °C to 37.0 °C with a mean of 6.5 mm daily rainfall, with 12 days of rain (of > 1 mm) over the total 40-day period . |
| Location | Our study site was adjacent to Murrua Trail, Murrua Side Trail, and Gibberagong Trail within Ku- ring- gai Chase National Park, Sydney, Australia (33°41'33"S, 151°08'44"E) |
| Access & import/export | Research was conducted under a NSW Government Department of Planning, Industry and Environment Scientific Licence (Biodiversity Conservation Act 2016), Licence number: SL102186. |
| Disturbance | There was no disturbance caused by the study. |

# Reporting for specific materials, systems and methods

We require information from authors about some types of materials, experimental systems and methods used in many studies. Here, indicate whether each material, system or method listed is relevant to your study. If you are not sure if a list item applies to your research, read the appropriate section before selecting a response.

## Materials & experimental systems

| n/a | Involved in the study |
|---|---|
| ☒ | ☐ Antibodies |
| ☒ | ☐ Eukaryotic cell lines |
| ☒ | ☐ Palaeontology and archaeology |
| ☐ | ☒ Animals and other organisms |
| ☒ | ☐ Clinical data |
| ☒ | ☐ Dual use research of concern |
| ☒ | ☐ Plants |

## Methods

| n/a | Involved in the study |
|---|---|
| ☒ | ☐ ChIP-seq |
| ☒ | ☐ Flow cytometry |
| ☒ | ☐ MRI-based neuroimaging |

# Animals and other research organisms

Policy information about studies involving animals; ARRIVE guidelines recommended for reporting animal research, and Sex and Gender in Research

| | |
|---|---|
| Laboratory animals | The study did not involve laboratory animals. |

| Wild animals | Swamp wallabies (Wallabia bicolor) were observed using non-invasive motion-triggered infra-red trail cameras on plots. The animals were not captured or handled in any way. The age of wild swamp wallabies observed could not be determined. |
| Reporting on sex | This information was not collected. |
| Field-collected samples | The study did not involve samples collected from the field. |
| Ethics oversight | Animal ethics approval was granted by the University of Sydney's Animal Ethics Committee (protocol number 2022/2196). |

Note that full information on the approval of the study protocol must also be provided in the manuscript.

