## [Peer Review File · Nature Ecology & Evolution]

Peer Review Information

Journal: Nature Ecology & Evolution

Manuscript Title: Olfactory misinformation provides refuge to palatable plants from mammalian browsing

Corresponding author name(s): Patrick B. Finnerty, Clare McArthur

Editorial Notes:

Transferred manuscripts (no peer review at Nature XX)	This manuscript has been previously reviewed at another journal and was considered suitable for publication without further review at Nature XX.
Redactions – transferred manuscripts (mention of previous referee reports from elsewhere)	This manuscript has been previously reviewed at another journal. This document only contains reviewer comments, rebuttal and decision letters for versions considered at Nature XX. Mentions of prior referee reports have been redacted
Redactions – transferred manuscripts (mention of the other journal)	This manuscript has been previously reviewed at another journal. This document only contains reviewer comments, rebuttal and decision letters for versions considered at Nature XX. Mentions of the other journal have been redacted.
Redactions – unpublished data	Parts of this Peer Review File have been redacted as indicated to maintain the confidentiality of unpublished data.
Redactions – confidential patient information	Parts of this Peer Review File have been redacted as indicated to maintain patient confidentiality.
Redactions – published data	Parts of this Peer Review File have been redacted as indicated to remove third-party material.
Redactions – reviewer opt-out	Parts of this Peer Review File have been redacted as indicated as we could not obtain permission to publish the reports of reviewer no. XX.
Reviewer comments in marked-up manuscript	In their review of the [first/second/third/...] version of this manuscript, reviewer no. XX added their comments to the manuscript file. These comments, excluding minor textual revisions, have been copied into this Peer Review File.

Reviewer Comments & Decisions:

Decision Letter, initial version:

31st October 2023

Dear Mr Finnerty,

Your manuscript entitled "Olfactory misinformation provides refuge to palatable plants from mammalian browsing" has now been seen by three reviewers, whose comments are copied below. The reviewers appreciate the contribution, but have raised several concerns which we would like to see addressed in a revised manuscript before we can reach a final decision regarding publication in Nature Ecology & Evolution. We therefore invite you to revise your manuscript taking into account all reviewer and editor comments. Please highlight all changes in the manuscript text file.

Editorially, we would like to see the paper toned down in several ways. First, the management implications should be discussed more cautiously, given that scalability and effectiveness in natural settings could be limited. Second, please remove statements like "Our findings are a huge leap forward" and better acknowledge that the results are system-specific (to address Reviewer 3's concerns about limited generality).

* If you have not done so already please begin to revise your manuscript so that it conforms to our Brief Communication format instructions at <http://www.nature.com/natecolevol/info/final-submission>. Refer also to any guidelines provided in this letter.

2Please use the link below to submit your revised manuscript and related files:

[REDACTED]

Nature Ecology & Evolution is committed to improving transparency in authorship. As part of our efforts in this direction, we are now requesting that all authors identified as 'corresponding author' on published papers create and link their Open Researcher and Contributor Identifier (ORCID) with their account on the Manuscript Tracking System (MTS), prior to acceptance. ORCID helps the scientific community achieve unambiguous attribution of all scholarly contributions. You can create and link your ORCID from the home page of the MTS by clicking on 'Modify my Springer Nature account'. For more information please visit www.springernature.com/orcid.

[REDACTED]

Reviewers' comments:

Reviewer #1 (Remarks to the Author):

Review of NATECOLEVOL-23092136 "Olfactory misinformation provides refuge to palatable plants from mammalian browsing."

The authors document the use of virtual neighbors to be as effective as using unpalatable plants to deter a mammalian herbivore from browsing on an economically and ecologically important plant. The authors describe their methodology for determining important volatile organic compounds (VOCs) present in unpalatable plants in order to make a virtual neighbor, as well as the dispenser for the virtual neighbor. Their methodology and results were novel and will be of interest to a wide range of ecologists, and land managers. The study was well designed and the methods adequately described. I

2just have a few suggestions for the reporting of the results.

Most of the results are in supplementary data. Unless there is a space limitation on the manuscript, I would recommend moving some of the results from supplementary figures and tables into the manuscript. Particularly Supplementary figure 2 and 3 and table 1 and table 2ii.

The results of VOC analysis of *Boronia pinnata* are not reported in detail. Were there differences in between the plants sampled in March 2021 and April 2022? How about differences between the "wild" plants sampled in 2021 and 2022 and the greenhouse raised plants? This data would be helpful to people wanting to know how much VOC emission varies due to site, season, and rearing conditions. These results could be put into a supplementary table.

Line 135 when it is reported that there was no difference in wallaby density across treatments report the wallaby density.
Knowing the actual animal density will be of interest to readers.

Lines 333-334 states that the proportion of *E. punctata* biomass consumed after first browsing visit was quantified. How was it quantified? Based on the camera data? Line 133 reports that a mean of $97 \pm 0.01\%$ of the seedling biomass was consumed. That seems like a very specific amount if the amount quantified was based on camera data.

Lines 354-357 describes the analysis of the effect of treatment and background wallaby activity on the proportion of palatable seedling browsed at first browsing visit but I cannot find where those results are reported.

Other questions/concerns

Line 341-342 state that any plots where wallabies were not recorded on camera traps either during the pre-trial period or main trial were removed from analysis. However it was not stated anywhere if any plots were removed from analysis.

-Michele M. Skopec

Reviewer #2 (Remarks to the Author):

This was a nicely written paper on an interesting topic, with an eye towards application. There are relatively few associational effects papers that so thoroughly investigate plant chemistry as a mechanism underlying associational resistance. The methods must have required extensive and repeated development. I have just a couple main questions about the methods, which I think could use more contextualization.

L 69, 87: The authors describe a new statistical method to come up with informative VOC blends. L69 makes it clear this is a novel method, but L87 implies that it has use in other systems ('usually described in terms of VOC pairs'). I might have missed something but I can't find a mention of VOC

3pairs in the cited Webster et al. 2010. This is an interesting method, but it should be made clear if it has only been used in Eucalyptus.

L 222-259 I can see why using a computational method to determine the most informative set of compounds is very powerful, but I'm unclear on why the final concentrations of those compounds are not just based on the average proportions in the original data. Is there too much variation in the relative frequency of these 7 compounds across individuals of the species? Or is there another reason additional trials of variable pairwise proportions is required? More context would help for understanding how easy this would be to apply in other systems, and how it compares with previous efforts to identify informative VOCs.

L 126: Although the untreated control and procedural control are equivalent statistically, the effect size of the 'virtual neighborhood' should still be presented relative to the procedural control.

L269: The plants are stated to be the same starting biomass, and I am assuming this is how biomass loss was calculated (L334). Was the average dry weight determined through prior trials?

Reviewer #3 (Remarks to the Author):

This study provides fascinating evidence for the effects of "virtual plants", i.e. the presence of plant odours of unpalatable plants, in protecting palatable plants from browsing by a large herbivore, which can be considered as associational plant refuge. This method could be applied to protect crop plants from unwanted herbivory. The authors used an elegant set of experiments with target plants without neighbours, with real neighbours of an unpalatable plant or with different volatile combinations of this unpalatable plant. Thereby, they considered both the identity and the exact proportion of volatiles in their information content. For emissions of the exact proportions they did important control measurements. They could demonstrate that the exact proportion also plays a large role in reducing foraging by wallabies, while flipped proportions or uninformative volatile mixtures did not protect the target seedlings from browsing.

Overall, the manuscript is very well written with a nice narrative. I just wonder how specific the findings may be to this particular location. Would the authors expect similar findings in different areas where the wallabies and respective plant species occur or could that be a learned response based on previous experience that is very specific for just this one colony of wallabies? Also, plant species may show different combinations of odours at different locations/in different populations based on intraspecific variation. In other words, while the evidence for this particular study system is convincing, I would be careful with generalising the findings.

Specific comments to the authors:

Line 92-93; 218ff and Figure S2: the names of the terpenoids (all chemicals) should be written in lower case letters.

Line 134: This day / night comparison comes as surprise. What is the relevance – or what was the underlying hypothesis – here?

4Line 156: Whether this approach is indeed directly transferable to ANY vertebrate or even invertebrate herbivore species can be doubted. Different species use different cues and for some a combination of visual and odour cues may be more important, for others even visual cues may be more crucial. Alternatively, just a single compound may be a very effective repellent, depending on the animal species. Thus, this sentence should be tuned down to some extent.

This entire paragraph (154ff) could be shortened, as some of the statements are redundant to statements already given in the text above.

Line 202-204: check sentence, something is awkward here.

Line 253: consider replacing "when developing *B. pinnata* odour profile" with "when measuring the odour profiles from *B. pinnata* plants".

What was the total number of wallaby individuals that were monitored? Please add this information.

I like the conceptual figure 1. The green and purple colours for the palatable target and unpalatable neighbour odours may be changed to colours that can be better differentiated.

Line 429: Is it really degraded "quality" or rather the presence of repellent odours, which do not necessarily affect the quality?

Figure 2 and Supplement Table 2: What does "x5" mean in the legend?

The supplement figures give a nice and helpful impression about the set-up.

*****END*****

Author Rebuttal to Initial comments

Response to Referees – Nature Ecology & Evolution Manuscript NATECOLEVOL-23092136

Senior Editor			
Senior editor Suggestion	Author Response	Line #	New Text
Editorially, we would like to see the paper toned down in several ways. First, the management implications should be discussed more cautiously, given that scalability and effectiveness in natural settings could be limited.	We have toned down our discussion of potential management implications throughout our manuscript. We agree with your comment, as no management strategy is ever effective in all scenarios. And we agree that scalability matters. But there will be scales where we think the strategy should be possible (for example threatened plant species with small local distributions) even in natural settings (as in our study).	159 - 163	"Herbivore browsing damage varies in detail and context globally: different plants, different herbivores, different landscapes. However, irrespective of the context, the logical approach used to define the putatively informative compounds of plant species is likely transferable to many mammalian (or potentially invertebrate) herbivores that rely primarily on plant odour information to forage. Consequently, using similar olfactory misinformation tactics, virtual neighbourhoods represent a new approach that has the potential to..."
Second, please remove statements like "Our findings are a huge leap forward" and better acknowledge that the results are system-specific (to address Reviewer 3's concerns about limited generality).	We have removed this statement. However, we emphasise that there is much about our approach and results that is not system-specific. Herbivores (not just wallabies) avoid unpalatable plants (not just Boronia pinnata). Odour is the first cue to enable them (not just wallabies) to do so. We demonstrate that our approach to detect and quantify informative VOCs has worked, and that it alters herbivore foraging behaviour as predicted. These are the key features that we argue make our results and conclusions so exciting and broadly applicable to many ecosystems and plant-herbivore systems around the world.	147 - 149	"Our findings provide an important step forward in improving our understanding of both fundamental and applied mammalian behavioural ecology, providing new insight into the ways in which mammalian herbivores detect and respond to the world around them."
		149 - 151	"We argue that our approach to detect and quantify informative VOCs has the potential to be applied more broadly to develop targeted virtual plant neighbours specific to herbivores in other systems."
Reviewer #1			
Reviewer Suggestion	Author Response	Line #	New Text
Most of the results are in supplementary data. Unless there is a space limitation on the manuscript, I would recommend moving some of the results from supplementary figures and tables into the manuscript. Particularly Supplementary figure 2 and 3 and table 1 and table 2ii.	In following space/page limits for a 'brief communication' article type for Nature Ecology & Evolution, we report most of our results (including figures and tables) in supplementary materials. However, we agree with your comment and so have moved	94 and 413 - 422	Reference to Figure 2, and the location of Figure 2 in the revised manuscript.
		101 and 444 - 447	Reference to Table 1, and the location of Table 1 in the revised manuscript.
		109 and 423 - 435	Reference to Figure 3, and the location of Figure 3 in the revised manuscript.

	Supplementary figure 2 and 3 and table 1 and table 2ii into the main manuscript. We are happy to comply with the editor's decision either way, about whether these figures and tables are appropriate for this article type and can remain in the main text.	125 and 448 - 450	Reference to Table 2, and the location of Table 2 in the revised manuscript.
The results of VOC analysis of Boronia pinnata are not reported in detail. Were there differences in between the plants sampled in March 2021 and April 2022? How about differences between the "wild" plants sampled in 2021 and 2022 and the greenhouse raised plants? This data would be helpful to people wanting to know how much VOC emission varies due to site, season, and rearing conditions. These results could be put into a supplementary table.	We now report results of VOC analysis of Boronia pinnata in detail as requested. To do this, we have included a data matrix of VOC peak areas, with associated metadata (Supplementary data matrix 1). This matrix allows comparison of wild B. pinnata odour profiles identified across both sampling bouts. We have also now included a synopsis (Supplementary Note 1) summarising these differences. For the nursery plants, we focused only on the emission rates of the few informative compounds identified from wild B. pinnata sampling. We took this approach to be efficient, because it takes a huge (unnecessary) effort to identify all VOCs matched to peak areas in the complete odour profile. We also used a different sampling method. As we did not identify all VOCs in the odour profile for these plants, we cannot provide that information. However, we can compare VOC paired proportions across of the seven selected informative VOC pairs between wild and nursery raised B. pinnata. We now provide this data as Supplementary Figure 2. VOC paired proportions are similar across both wild and nursery raised B. pinnata.	85	"Supplementary Note 1"
		87 - 88	"see Supplementary data matrix 1 and Supplementary Figure 2 for a comparison of the odour profiles between sampling bouts"
		212 - 213	"see Supplementary Note 1, Supplementary data matrix 1 and Supplementary Figure 2 for a comparison of the odour profiles between sampling bouts"
Line 135 when it is reported that there was no difference in wallaby density across treatments	Yes, actual animal density may be of interest to readers, but we do not have that data and	136	"Background wallaby activity did not differ across treatments"

report the wallaby density. Knowing the actual animal density will be of interest to readers.	it is not essential for our study. We did not quantify absolute wallaby density. Rather, we obtained a measure of wallaby activity, calculated as the total number of wallaby visits made to each plot in the 14-day pre-trial period (before we ran our experiment). This activity measure is more ecologically relevant for our purposes, because it should directly affect how quickly a plot is visited. We have changed the wording from wallaby density to wallaby activity so as not to confuse readers.	323 - 325	"...calculate a score of background wallaby activity per patch (background wallaby activity score = the number of wallabies recorded at a treatment site during the pre-trial period)."
Lines 333-334 states that the proportion of E. punctata biomass consumed after first browsing visit was quantified. How was it quantified? Based on the camera data? Line 133 reports that a mean of 97±0.01% of the seedling biomass was consumed. That seems like a very specific amount if the amount quantified was based on camera data.	Yes, this appears to be a very specific amount, but that is because it is summary data (means of 82 seedlings). We estimated the % of foliage consumed from each of 82 seedlings as seen on camera using a visual estimate, with % intervals of 0, 25, 50, 75 and 100% eaten. Most (82 of 87) seedlings had 100% of foliage consumed (easy to see on camera). Only 7 seedlings were partially consumed but these all were estimated to be 75% consumed (also easy to see). The average of all these browse values is 98% and the s.e. is 1%. However, to avoid confusion, we have changed the wording from "if E. punctata seedlings were browsed, wallabies ate almost all biomass (mean 97.0 ± 0.01 %)." To "if E. punctata seedlings were browsed, wallabies generally ate all the foliage (75 of 82 seedlings) or most foliage (7 of 82 seedlings)."	134 - 136	"If E. punctata seedlings were browsed, wallabies generally ate all the foliage (75 of 82 seedlings) or most foliage (7 of 82 seedlings)"
Lines 354-357 describes the analysis of the effect of treatment and background wallaby activity on the proportion of palatable seedling browsed at first browsing visit but I cannot find where those results are reported.	Sorry – we have caused confusion here. We originally had this analysis in mind in case there was variation in amount of foliage browsed. However, it turned out that wallabies generally ate all the foliage (75 of 82 seedlings) or most foliage (7 of 82 seedlings) [see response to preceding comment], and the latter were distributed across 4 of the 6 treatments. This meant there was insufficient variation for there to be any treatment effect	NA	NA

	or wallaby activity effect, and so the beta regression was subsequently superfluous. We have now removed any reference to this analysis.		
Line 341-342 state that any plots where wallabies were not recorded on camera traps either during the pre-trial period or main trial were removed from analysis. However it was not stated anywhere if any plots were removed from analysis.	Thank you for pointing this out. This line was written with the intention of doing so (assuming it happened). When the data came in, we ended up with wallabies visiting all plots at some stage and hence data for all plots were valid and none were removed. We have therefore removed these lines from the manuscript.	NA	NA
Reviewer #2			
Reviewer Suggestion	Author Response	Line #	New Text
L 69, 87: The authors describe a new statistical method to come up with informative VOC blends. L69 makes it clear this is a novel method, but L87 implies that it has use in other systems ('usually described in terms of VOC pairs'). I might have missed something but I can't find a mention of VOC pairs in the cited Webster et al. 2010. This is an interesting method, but it should be made clear if it has only been used in Eucalyptus.	Apologies, we referenced the wrong Webster et al. paper from 2010. We have updated the revised manuscript with the correct Webster et al. 2010 reference, which does mention VOC pairs: 'Although the quantities of volatiles produced by V. faba showed large between plant and diurnal variation, correlations between quantities of compounds indicated that the ratios of certain pairs of volatiles were very consistent. This suggests that there is a host-characteristic cue available to A. fabae in the form of ratios of volatiles.' We also stress that our novel approach is not a statistical method. Rather it is a logical approach, and it is not system specific. We are not sure what the referee means by "has only been used in Eucalyptus". In the 2022 paper describing the concept and method for the first time, we used two tree species - one Eucalyptus and one Corymbia species - to detect and define putative informative VOCs. But in this current paper, now, we demonstrate the effectiveness and ecological	379 - 380	"Webster, B., Gezan, S., Bruce, T., Hardie, J. & Pickett, J. Phytochemistry 71, 81-89, (2010)."

	significance of our approach with a completely different Genus (Boronia) and life form (shrub).		
L 222-259 I can see why using a computational method to determine the most informative set of compounds is very powerful, but I'm unclear on why the final concentrations of those compounds are not just based on the average proportions in the original data. Is there too much variation in the relative frequency of these 7 compounds across individuals of the species? Or is there another reason additional trials of variable pairwise proportions is required? More context would help for understanding how easy this would be to apply in other systems, and how it compares with previous efforts to identify informative VOCs.	We did calculate the final proportions based on the average proportions in the original data — apart from minor adjustment. The adjustment was made because we deployed nursery plants for our real plant treatment in the manipulative experiment, and so we wanted to ensure emissions matched those from these plants. Wild and nursery emissions were similar (data now provided in Supplementary Figure 2 for comparison), although the averages were slightly different hence the adjustment. We also adjusted the absolute volume to ensure the emission rate (not just relative amounts) matched the nursery plants. We have clarified this in methods sections A.2 – A.3, providing the context and reason for analysing both wild and nursery plants. So yes, the approach is easy to use and could be applied to other systems. In reply to the comment “how [the method] compares with previous efforts to identify informative VOCs”, we have not gone into detail here because we discuss this extensively and thoroughly in Orlando et al 2022. It seems redundant (and would take up a lot of space) to repeat it again here.	214 – 286: Methods sections A.2 and A.3	Clarified in Methods sections A.2 and A.3
L 126: Although the untreated control and procedural control are equivalent statistically, the effect size of the 'virtual neighborhood' should still be presented relative to the procedural control.	We refer readers to see Supplementary Table. 1 where pairwise comparisons between all treatments are reported, including 'virtual neighborhood' vs procedural control, as requested.	134	“Supplementary Table 1”
L269: The plants are stated to be the same starting biomass, and I am assuming this is how biomass	Line 269 is referring to B. pinnata yet line 334 is referring to E. punctata , so the reviewer is mixing up the two plant species in this	134 - 136	“If E. punctata seedlings were browsed, wallabies generally ate all the foliage (75 of 82 seedlings) or most foliage (7 of 82 seedlings)”

loss was calculated (L334). Was the average dry weight determined through prior trials?	comment. B. pinnata plants were weighed (for line 269) and so the term biomass is correct and we have left it in. However, as in our response to reviewer #1 above, for line 334, to avoid confusion, we have changed the wording from "If E. punctata seedlings were browsed, wallabies ate almost all biomass (mean $97.0 \pm 0.01\%$). To "If E. punctata seedlings were browsed, wallabies generally ate all the foliage (75 of 82 seedlings) or most foliage (7 of 82 seedlings)".	318 - 320	"If browsed, we estimated the percentage of foliage consumed from each of E. punctata seedlings as seen on camera using a visual estimate, with % intervals of 0, 25, 50, 75 and 100% eaten."
Reviewer #3			
Reviewer Suggestion	Author Response	Line #	New Text
Fust wonder how specific the findings may be to this particular location. Would the authors expect similar findings in different areas where the wallabies and respective plant species occur or could that be a learned response based on previous experience that is very specific for just this one colony of wallabies?	It is unclear what findings the reviewer is referring to here. However, in response, yes, wallabies, or any species, may learn about the particular plants in their environments. But there is no reason why the response of this herbivore to that species should be "colony" specific. Our herbivore and plant species co-occur across hundreds of kilometres in eastern Australia. However, none of this affects our argument that the approach we use to find informative compounds and use them to alter foraging is generalisable. We have altered to wording in our revised manuscript to make this clearer.	149 - 151 159 - 163	"We argue that our approach to detect and quantify informative VOCs has the potential to be applied more broadly to develop targeted virtual plant neighbours specific to herbivores in other systems." "However, irrespective of the context, the logical approach used to define the putatively informative compounds of plant species is likely transferable to many mammalian (or potentially invertebrate) herbivores that rely primarily on plant odour information to forage. Consequently, using similar olfactory misinformation tactics, virtual neighbourhoods represent a new approach that has the potential to be harnessed..."
Also, plant species may show different combinations of odours at different locations/in different populations based on intraspecific variation. In other words, while the evidence for this particular study system is convincing, I would be careful with generalising the findings.	When saying be careful over generalising, we think the reviewer is referring to the specific set of VOCs in the specific proportions we used in our trials. We agree. However, as we argue above, our approach could be used to define the putatively informative compounds of plant species in different locations, or across different populations, and so is generalisable. So although the actual VOCs that are informative across locations may vary, the principle remains the same. Our goal here was not to provide the specific recipe for managers to reduce browsing by swamp	149 - 151 159 - 162	"We argue that our approach to detect and quantify informative VOCs has the potential to be applied more broadly to develop targeted virtual plant neighbours specific to herbivores in other systems." "However, irrespective of the context, the logical approach used to define the putatively informative compounds of plant species is likely transferable to many mammalian (or potentially invertebrate) herbivores that rely primarily on plant odour information to forage."

	wallabies. Rather, our goal was to demonstrate that the principle works in practice, and we have achieved this with our results.		
Line 92-93; 218ff and Figure S2: the names of the terpenoids (all chemicals) should be written in lower case letters.	We have made the required changes.	95, Figure 2: 413 – 422, Table 1: 444 – 447, Supplementary Figure 2	All chemical changed to lower case.
Line 134: This day / night comparison comes as surprise. What is the relevance – or what was the underlying hypothesis – here?	We included this comparison to provide an underlying context to wallaby foraging patterns. When foraging at night, odour cues should provide more information than visual cues as information. However, we have already reported this type of data in a previous paper. It does not add much here, and so we have removed the day/night comparison from our manuscript to avoid distracting the reader.	NA	NA
Line 156: Whether this approach is indeed directly transferable to ANY vertebrate or even invertebrate herbivore species can be doubted. Different species use different cues and for some a combination of visual and odour cues may be more important, for others even visual cues may be more crucial. Alternatively, just a single compound may be a very effective repellent, depending on the animal species. Thus, this sentence should be tuned down to some extent.	We agree that many animals use combinations of visual and odour cues to inform foraging decisions, and that for some animals, visual cues may be the driving mechanism. However, the point we were arguing was that this approach could be applied to any animal that uses olfaction as the first or key sense informing foraging decisions. We have altered the wording in our revised manuscript to hopefully make this point clearer. We also agree that a single compound could be an effective repellent. However, plant species rarely have a single compound that is unique to that species alone; VOC combinations identifying species are the norm. None of this detracts from our findings or conclusions.	159 - 162	"However, irrespective of the context, the logical approach used to define the putatively informative compounds of plant species is likely transferable to many mammalian (or potentially invertebrate) herbivores that rely primarily on plant odour information to forage."

This entire paragraph (154ff) could be shortened, as some of the statements are redundant to statements already given in the text above.	Agreed, this paragraph has been shortened.	158 - 165	"Herbivore browsing damage varies in detail and context globally: different plants, different herbivores, different landscapes. However, irrespective of the context, the logical approach used to define the putatively informative compounds of plant species is likely transferable to many mammalian (or potentially invertebrate) herbivores that rely primarily on plant odour information to forage. Consequently, using similar olfactory misinformation tactics, virtual neighbourhoods represent a new approach that has the potential to be harnessed as a benign, non-lethal, cost-effective, and novel tool for reducing problem herbivory in conservation (e.g., threatened plant species) and management (e.g., forestry and agriculture) globally."
Line 202-204: check sentence, something is awkward here.	We have altered the wording here.	202 - 204	"Blank samples (n = 7) were run in conjunction with all analyses; the upper 95% confidence interval of the mean blank value was subtracted from all samples"
Line 253: consider replacing "when developing B. pinnata odour profile" with "when measuring the odour profiles from B. pinnata plants".	Wording has been changed as suggested.	243 - 245	"Compounds were identified after GC-MS analysis using the same protocol used when measuring the odour profiles from B. pinnata plants"
What was the total number of wallaby individuals that were monitored? Please add this information.	As we stated above in our response to reviewer #1, we did not quantify absolute wallaby density. Rather, we obtained a measure of wallaby activity. We can therefore not add this information and have updated the wording in our manuscript.	136	"Background wallaby activity did not differ across treatments"
		323 - 325	"...calculate a score of background wallaby activity per patch (background wallaby activity score = the number of wallabies recorded at a treatment site during the pre-trial period)."
		328 - 329	"to test whether there was a difference in background wallaby activity (dependent variable) between treatment sites."
		331 - 332	"as a function of treatment and background wallaby activity"
I like the conceptual figure 1. The green and purple colours for the palatable target and unpalatable neighbour odours may be changed to colours that can be better differentiated.	We have changed the colours here to red and blue as they offer the highest level of distinction and fit within a colour blind-friendly palette. Colours have also been made bolder and more vibrant.	401 - 412	Figure 1

Line 429: Is it really degraded "quality" or rather the presence of repellent odours, which do not necessarily affect the quality?	Yes – the presence of unpalatable plants (confirmed by the fact wallabies do not eat them even if they are right up close (hence not "repelled" in that sense)) means that the net value of the patch is degraded by their presence. Odours are associated with consequences and hence become a cue. Odours from unpalatable plants are avoided because the message they send to herbivores is that they represent a low quality plant, not because they smell bad ("repel"). We were deliberately avoiding the use of the word repellent, as these odours are informative about unpalatable plants.	NA	NA
Figure 2 and Supplement Table 2: What does "x5" mean in the legend?	x 5 relates to the number of 'units' placed out around E. punctata seedlings. We have removed x 5 from Figure 2 and Table 2 (which was previously Supp Table 2) to avoid confusion.	401: Figure 1 legend, 4483: Table 2	NA
The supplement figures give a nice and helpful impression about the set-up.	Thank you, we appreciate this comment.	NA	NA

Decision Letter, first revision:

29th November 2023

Dear Dr. Finnerty,

Thank you for submitting your revised manuscript "Olfactory misinformation provides refuge to palatable plants from mammalian browsing" (NATECOLEVOL-23092136A). It has now been seen again by the original reviewers and their comments are below. The reviewers find that the paper has improved in revision, and therefore we'll be happy in principle to publish it in Nature Ecology & Evolution, pending minor revisions to satisfy the reviewers' final requests and to comply with our editorial and formatting guidelines.

The current version of your manuscript is in a PDF format; please email us a copy of the file in an editable format (Microsoft Word or LaTeX)-- we can not proceed with PDFs at this stage.

14We will then perform detailed checks on your paper and will send you a checklist detailing our editorial and formatting requirements in about one to two weeks. Please do not upload the final materials and make any revisions until you receive this additional information from us.

[REDACTED]

Reviewer #1 (Remarks to the Author):

I feel that the authors have adequately addressed not only my concerns but also the concerns of the other reviewers and the senior editor. As someone who studies endangered plants being eaten by herbivores I feel that the authors did not overstate the management implications for their study in the original manuscript, however their toned down management implications still get the point across that virtual neighbors could be a valuable tool for land managers dealing with nuisance herbivores.

Our ref: NATECOLEVOL-23092136A

8th December 2023

Dear Dr. Finnerty,

Thank you for your patience as we've prepared the guidelines for final submission of your Nature Ecology & Evolution manuscript, "Olfactory misinformation provides refuge to palatable plants from mammalian browsing" (NATECOLEVOL-23092136A). Please carefully follow the step-by-step instructions provided in the attached file, and add a response in each row of the table to indicate the changes that you have made. Please also check and comment on any additional marked-up edits we have proposed within the text. Ensuring that each point is addressed will help to ensure that your revised manuscript can be swiftly handed over to our production team.

We would like to start working on your revised paper, with all of the requested files and forms, as soon as possible (preferably within two weeks). Please get in contact with us immediately if you anticipate it taking more than two weeks to submit these revised files.

15When you upload your final materials, please include a point-by-point response to any remaining reviewer comments.

In recognition of the time and expertise our reviewers provide to Nature Ecology & Evolution's editorial process, we would like to formally acknowledge their contribution to the external peer review of your manuscript entitled "Olfactory misinformation provides refuge to palatable plants from mammalian browsing". For those reviewers who give their assent, we will be publishing their names alongside the published article.

Nature Ecology & Evolution offers a Transparent Peer Review option for new original research manuscripts submitted after December 1st, 2019. As part of this initiative, we encourage our authors to support increased transparency into the peer review process by agreeing to have the reviewer comments, author rebuttal letters, and editorial decision letters published as a Supplementary item. When you submit your final files please clearly state in your cover letter whether or not you would like to participate in this initiative. Please note that failure to state your preference will result in delays in accepting your manuscript for publication.

Cover suggestions

We welcome submissions of artwork for consideration for our cover. For more information, please see our https://www.nature.com/documents/Nature_covers_author_guide.pdf guide for cover artwork.

Nature Ecology & Evolution has now transitioned to a unified Rights Collection system which will allow our Author Services team to quickly and easily collect the rights and permissions required to publish your work. Approximately 10 days after your paper is formally accepted, you will receive an email in providing you with a link to complete the grant of rights. If your paper is eligible for Open Access, our Author Services team will also be in touch regarding any additional information that may be required to arrange payment for your article.

Please note that *Nature Ecology & Evolution* is a Transformative Journal (TJ). Authors may publish their research with us through the traditional subscription access route or make their paper immediately open access through payment of an article-processing charge (APC). Authors will not be

16required to make a final decision about access to their article until it has been accepted. [Find out more about Transformative Journals](https://www.springernature.com/gp/open-research/transformative-journals)

Authors may need to take specific actions to achieve [compliance with funder and institutional open access mandates](https://www.springernature.com/gp/open-research/funding/policy-compliance-faqs). If your research is supported by a funder that requires immediate open access (e.g. according to [Plan S principles](https://www.springernature.com/gp/open-research/plan-s-compliance)) then you should select the gold OA route, and we will direct you to the compliant route where possible. For authors selecting the subscription publication route, the journal's standard licensing terms will need to be accepted, including [self-archiving and license to publish](https://www.nature.com/nature-portfolio/editorial-policies/self-archiving-and-license-to-publish). Those licensing terms will supersede any other terms that the author or any third party may assert apply to any version of the manuscript.

Please use the following link for uploading these materials:
[REDACTED]

[REDACTED]

Reviewer #1:

Remarks to the Author:

I feel that the authors have adequately addressed not only my concerns but also the concerns of the other reviewers and the senior editor. As someone who studies endangered plants being eaten by herbivores I feel that the authors did not overstate the management implications for their study in the original manuscript, however their toned down management implications still get the point across that virtual neighbors could be a valuable tool for land managers dealing with nuisance herbivores.

Final Decision Letter:

11th January 2024

Dear Mr Finnerty,

We are pleased to inform you that your Brief Communication entitled "Olfactory misinformation provides refuge to palatable plants from mammalian browsing", has now been accepted for publication in Nature Ecology & Evolution.

Over the next few weeks, your paper will be copyedited to ensure that it conforms to Nature Ecology and Evolution style. Once your paper is typeset, you will receive an email with a link to choose the appropriate publishing options for your paper and our Author Services team will be in touch regarding any additional information that may be required

Due to the importance of these deadlines, we ask you please us know now whether you will be difficult to contact over the next month. If this is the case, we ask you provide us with the contact information (email, phone and fax) of someone who will be able to check the proofs on your behalf, and who will be available to address any last-minute problems . Once your paper has been scheduled for online publication, the Nature press office will be in touch to confirm the details.

Acceptance of your manuscript is conditional on all authors' agreement with our publication policies (see www.nature.com/authors/policies/index.html). In particular your manuscript must not be published elsewhere and there must be no announcement of the work to any media outlet until the publication date (the day on which it is uploaded onto our web site).

Please note that *Nature Ecology & Evolution* is a Transformative Journal (TJ). Authors may publish their research with us through the traditional subscription access route or make their paper immediately open access through payment of an article-processing charge (APC). Authors will not be required to make a final decision about access to their article until it has been accepted. [Find out more about Transformative Journals](https://www.springernature.com/gp/open-research/transformative-journals)

Authors may need to take specific actions to achieve [compliance with funder and institutional open access mandates](https://www.springernature.com/gp/open-research/funding/policy-compliance-faqs). If your research is supported by a funder that requires immediate open access (e.g. according to [Plan S principles](https://www.springernature.com/gp/open-research/plan-s-compliance)) then you should select the gold OA route, and we will direct you to the compliant route where possible. For authors selecting the subscription publication route, the journal's standard licensing

18terms will need to be accepted, including <https://www.nature.com/nature-portfolio/editorial-policies/self-archiving-and-license-to-publish>. Those licensing terms will supersede any other terms that the author or any third party may assert apply to any version of the manuscript.

We welcome the submission of potential cover material (including a short caption of around 40 words) related to your manuscript; suggestions should be sent to Nature Ecology & Evolution as electronic files (the image should be 300 dpi at 210 x 297 mm in either TIFF or JPEG format). Please note that such pictures should be selected more for their aesthetic appeal than for their scientific content, and that colour images work better than black and white or grayscale images. Please do not try to design a cover with the Nature Ecology & Evolution logo etc., and please do not submit composites of images related to your work. I am sure you will understand that we cannot make any promise as to whether any of your suggestions might be selected for the cover of the journal.

You can generate the link yourself when you receive your article DOI by entering it here: <http://authors.springernature.com/share>.

[REDACTED]

P.S. Click on the following link if you would like to recommend Nature Ecology & Evolution to your librarian <http://www.nature.com/subscriptions/recommend.html#forms>** Visit the Springer Nature Editorial and Publishing website at http://editorial-jobs.springernature.com?utm_source=ejp_NEcoE_email&utm_medium=ejp_NEcoE_email&utm_campaign=ejp_NEcoE for more information about our career opportunities. If you have any questions please click [here](mailto:editorial.publishing.jobs@springernature.com).**